# Hyperspectral Imaging for Glioblastoma Surgery: Improving Tumor Identification Using a Deep Spectral-Spatial Approach

**DOI:** 10.3390/s20236955

**Published:** 2020-12-05

**Authors:** Francesca Manni, Fons van der Sommen, Himar Fabelo, Svitlana Zinger, Caifeng Shan, Erik Edström, Adrian Elmi-Terander, Samuel Ortega, Gustavo Marrero Callicó, Peter H. N. de With

**Affiliations:** 1Department of Electrical Engineering, Eindhoven University of Technology, 5600 MB Eindhoven, The Netherlands; fvdsommen@tue.nl (F.v.d.S.); s.zinger@tue.nl (S.Z.); P.H.N.de.With@tue.nl (P.H.N.d.W.); 2Institute for Applied Microelectronics (IUMA), University of Las Palmas de Gran Canaria (ULPGC), 35017 Las Palmas de Gran Canaria, Spain; hfabelo@iuma.ulpgc.es (H.F.); sortega@iuma.ulpgc.es (S.O.); gustavo@iuma.ulpgc.es (G.M.C.); 3College of Electrical Engineering and Automation, Shandong University of Science and Technology, Qingdao 266590, China; caifeng.shan@gmail.com; 4Department of Neurosurgery, Karolinska University Hospital and Department of Clinical Neuroscience, Karolinska Institutet, SE-171 46 Stockholm, Sweden; erik.edstrom@sll.se (E.E.); adrian.elmi-terander@sll.se (A.E.-T.)

**Keywords:** hyperspectral imaging, glioblastoma, ant-colony-based band selection, tumor tissue classification, deep learning, brain imaging

## Abstract

The primary treatment for malignant brain tumors is surgical resection. While gross total resection improves the prognosis, a supratotal resection may result in neurological deficits. On the other hand, accurate intraoperative identification of the tumor boundaries may be very difficult, resulting in subtotal resections. Histological examination of biopsies can be used repeatedly to help achieve gross total resection but this is not practically feasible due to the turn-around time of the tissue analysis. Therefore, intraoperative techniques to recognize tissue types are investigated to expedite the clinical workflow for tumor resection and improve outcome by aiding in the identification and removal of the malignant lesion. Hyperspectral imaging (HSI) is an optical imaging technique with the power of extracting additional information from the imaged tissue. Because HSI images cannot be visually assessed by human observers, we instead exploit artificial intelligence techniques and leverage a Convolutional Neural Network (CNN) to investigate the potential of HSI in twelve in vivo specimens. The proposed framework consists of a 3D–2D hybrid CNN-based approach to create a joint extraction of spectral and spatial information from hyperspectral images. A comparison study was conducted exploiting a 2D CNN, a 1D DNN and two conventional classification methods (SVM, and the SVM classifier combined with the 3D–2D hybrid CNN) to validate the proposed network. An overall accuracy of 80% was found when tumor, healthy tissue and blood vessels were classified, clearly outperforming the state-of-the-art approaches. These results can serve as a basis for brain tumor classification using HSI, and may open future avenues for image-guided neurosurgical applications.

## 1. Introduction

Glioblastoma multiforme (GBM) is one of the most aggressive forms of malignant gliomas [1]. Surgery aims to achieve gross total resection, which, when combined with radio- and chemotherapy, is associated with improved survival. During surgery, tumor margins are identified visually in the surgical microscope. However, GBM infiltrates the normal parenchyma and the likelihood of recurrence or residual tumor is very high when only visual delineation is used. Conversely, supratotal resection can cause severe neurological impairment [2]. Assistance in defining the tumor margins is provided by preoperative imaging, mainly magnetic resonance imaging (MRI). Intraoperative neurosurgical navigation techniques help the surgeon to define the intended resection. However, due to the brain-shift phenomenon, tumor boundaries change during the course of a surgery, yielding an inaccurate navigation [3]. Intraoperative imaging modalities such as ultrasound and MRI, as well as fluorescent tumor agents, may be used to provide up-to-date information. Intraoperative MRI has been suggested as a method to compensate for brain shift, but suffers the drawbacks of poor spatial resolution, high cost and time-consuming workflow [4]. In addition, it is not feasible to perform multiple intraoperative MRI cycles during a single surgery [4,5]. Intraoperative fluorescent agents can help to identify GBM tissue, but the definition of tumor margins remains difficult due to the infiltrative growth [6]. Histopathological examinations are used to verify tumor-free margins and total resection in oncological surgery. However, this is not applicable to glioma surgery. The turn-around time of tissue analysis is counted in hours and gliomas cannot be resected with a wide tumor-free margin, due to the risk of neurological impairment.

Thus, intraoperative and non-ionizing imaging techniques able to real-time identification of tumor tissue could considerably expedite the clinical workflow for tumor resection and improve outcomes by facilitating gross total resection.

Hyperspectral imaging (HSI) is a spectral-based imaging modality that acquires data in almost contiguous narrow spectral bands. HSI systems can collect hundreds of bands in specific parts of the electromagnetic spectrum, reconstructing a 3D cube which forms a set of 2D images, the so-called hyperspectral (HS) cube [7]. The HS cube combines conventional imaging and spectroscopy to capture the spatial and the spectral information, providing hundreds of spectral bands for each pixel in the 2D plane [8]. Each pixel contains an almost continuous spectrum to create a spectral signature representing radiance, reflectance and absorption, acting as a fingerprint (the so-called spectral signature) which reflects the chemical composition of that particular pixel [8]. The main advantage of this technology is that it only uses white light to acquire the wavelengths of interest, where an HS camera can scan through those wavelengths so that the spectral and spatial information can be detected in a non-invasive way without physical contact [9]. While originally employed in the remote sensing field, HSI has recently found use in the detection and diagnosis of diseases, in particular cancer, since cancer involves changes in the biochemical mechanism of the cells [10]. The changes in the cellular morphology and metabolism can be detected as changes in the absorption and reflectance of light within tissue and thus consequentially detected with the HSI camera. HSI in combination with cutting-e.g., machine learning and deep learning algorithms [9,10,11] has been used to detect prostate [12], breast [13,14], colon [11,15], oral, tongue [16,17,18,19,20,21], cervix [22], and skin cancer [23,24]. In this scenario, brain tumors are extremely difficult to identify with the naked eye, because they deeply infiltrate the healthy brain tissue [25]. Furthermore, it is crucial to preserve the surrounding healthy areas, where there is no redundancy and usually the resected tumor margins are less healthy tissue [26]. In GBM surgery, HSI has been applied to create an in vivo HS human-brain image database [25] and develop a framework for qualitative tumor margin detection [27,28], with the aim of providing a surgical tool that is capable of visualizing the parenchymal area of the brain and the tumor location. Since HSI allows the surgeon to observe what cannot be seen with the human eye, several challenging aspects should be considered. First, the HS data consist of a combination of spectral and spatial information, and it is crucial to jointly catch both types of information when performing a classification task. Second, the HS data are high-dimensional data, characterized by high redundancy on the amount of image bands, which may potentially degrade the information and decrease the classification accuracy. Third, HS data are heavy to compute, which is an important aspect for real-time use in surgery. To overcome these challenges, it is necessary to deploy processing algorithms able to reduce the dimensionality of the HS data without losing the relevant information and combine them with a classification tool able to exploit the spectral and spatial information [29]. Roy et al. proposed a fusion of 3D–2D CNN, a so-called Hybrid-SpectraNet (HybridSN), to jointly discriminate the features from both spectral and spatial information [30]. We have previously applied a modified version of the HybridSN, to assess colon cancer margins [15].

In this work, a hybrid deep learning-based framework is presented to quantitatively classify brain and tumor tissue on using an in vivo HS brain dataset. An automated band-selection algorithm was applied to reduce the computation time and provide insights about the relevant spectral bands towards a real-time HSI acquisition system [29]. Our framework demonstrates the feasibility for HSI-based classification of tumor tissue and highlights the differentiation between healthy brain tissue and blood vessels, all together. The framework provides a basis for aiding neurosurgeons in their critical surgical procedure during brain tumor surgery, while showing higher accuracy compared to the state-of-the-art approaches.

## 2. Materials and Methods

This section presents the HSI system used to acquire the in vivo HS brain cancer dataset, as well as the proposed framework for HS image pre-processing, band selection and brain tissue classification.

### 2.1. Intraoperative HS Image Acquisition System

An HSI acquisition system was designed to collect data for the creation of an in vivo HS human-brain database [25]. This acquisition system is described in detail in [31]. HS data were captured in the VNIR (visible and near-infrared) spectral range between 400 and 1000 nm. The system employed is a Hyperspec^®^ VNIR A-Series pushbroom camera (Headwall Photonics Inc., Fitchburg, MA, USA) able to obtain 826 bands with a spectral resolution of 2–3 nm and a sampling interval of 0.73 nm. The HS camera captures the complete spectral dimensions and only one spatial dimension of the scene in a single shot. For this reason, it is necessary for the scanning to apply a linear displacement stage. By sliding the camera with the scanning technique, the complete HS cube is obtained. The maximum spatial dimensions is 1004 × 826 pixels. The working distance is 40 cm, covering an maximum effective area of 129 × 230 mm. The pixel size is 128.7 μm. The HS acquisition system was installed at the University Hospital Doctor Negrin of Las Palmas de Gran Canaria (Spain) and the University Hospital of Southampton (UK) and was used to capture brain surface exposed to white light. The illumination system based on a 150-W QTH (quartz-tungsten-halogen) lamp is connected to a cold light emitter via a fiber-optical light guide. The cold light emitter ensures that the brain is not subjected to high temperatures produced by the QTH lamp in the exposed brain surface.

### 2.2. In Vivo Human-Brain HS Dataset

Within the context of the European project HELICoiD (HypErspectraL Imaging Cancer Detection) (Grant Agreement 618080), an in vivo human-brain HS database was collected, consisting of twenty-six images (*n* = 26) from sixteen adult patients [25]. This project had the main goal of demonstrating, as a proof-of-concept, that the use of HSI can be helpful for the identification and delineation of in-vivo human brain tumor boundaries in real time during neurosurgical operations. The study presented in this manuscript employs data from 16 patients collected in this project which are publicly available in [25]. Nine patients had a histopathologically confirmed Grade IV glioblastoma (GBM), while the remaining seven patients were either affected by other types of tumors, or affected by other pathologies that required a craniotomy. Next, a brief description of the procedure followed to acquire such data during the neurosurgical operations (Figure 1a) within the HELICoiD project execution is presented. More details can be found in [25]. During surgery, after craniotomy and dural opening, the neurosurgeon used preoperative imaging data to identify and mark normal brain and tumor tissue by placing rubber-ring markers (with an external diameter of 10 mm), as shown in Figure 1b. HS images were then captured during surgery and biopsies of the tissue inside the tumor marker were taken to confirm the presence of malignant tumor and determine its type and grade. Resected tissue was sent for histopathological analysis. In case of superficial tumors, HS images were acquired after the opening of the dura. Tumors located in a deep layer were recorded after beginning tumor resection. The labelling tool based on the Spectral Angle Mapper (SAM) algorithm was employed to label data as described in [25]. This tool was employed by the operating neurosurgeons to create the ground truth dataset. Reference pixels for the normal and tumor classes were selected in the image inside the ring markers and then pixels with similar spectrum in the image were selected to conform the ground truth for such classes (according to the neurosurgeons criteria). Additionally, the tumor marker is employed to identify the place where the tissue biopsy was performed for confirming the pathological diagnosis of the tumor. The SAM was then applied to the pixels previously selected and a threshold was set to find and select other pixels with almost the same spectral properties. Tumor pixels were labelled based on the biopsy assessment. Neurosurgeons labeled the normal tissue, blood vessels and background by visual inspection according to their experience. In previous works, the blood vessels class was called hyper-vascularized. However, since the labelled pixels in such class involves mainly blood vessels, in this work we redefined this term because hyper-vascularized can be confusing for medical readers. Figure 1b shows an example of the synthetic RGB image of brain tissue and Figure 1c the corresponding obtained ground-truth map, where green, red, blue and black pixels represent normal, tumor, blood vessels and background labeled samples, respectively. The background class includes tissue and materials other than the brain parenchyma, such as dura mater or surgical material, etc., exposed in the HS images. More details can be found in [25]. Figure 1d–g depict examples of gray-scale band representations for different wavelengths in the employed spectral range. The total numbers of labeled pixels for each class within the HS images are listed in Table 1. The technology adopted from Fabelo et al. [25], was not modified for performing new HS acquisitions in this study. The study and consent procedures were approved by the Comité Ético de Investigación Clínica-Comité de Ética en la Investigación (CEIC/CEI)of the University Hospital Doctor Negrin and the National Research Ethics Service (NRES) Committee South Central—Oxford C for the University Hospital of Southampton. Written consents were obtained from all participating patients.

### 2.3. HS Data Preprocessing and Band Selection

HS image preprocessing is a very important step due to the instrumentation noise and the curse of data dimensionality. The raw data acquired by the sensor are normalized for correction of the dark noise, using a dark and a white reference, following Equation (Equation 1):(1)$$I_{\mathrm{ref}}\left(p_{k}\right)=\frac{I_{\mathrm{raw}}\left(p_{k}\right)-I_{\mathrm{dark}}\left(p_{k}\right)}{I_{\mathrm{white}}\left(p_{k}\right)-I_{\mathrm{dark}}\left(p_{k}\right)}\;\;\mathrm{with}\;k\in \left[1,\ldots K\right],$$
where $I_{\mathrm{ref}}$ is the normalized reflectance value, $I_{\mathrm{raw}}$ is the diffuse reflectance value at the *k* pixel $p_{k}$, $I_{\mathrm{white}}$ represents the intensity value for the white reference plate and $I_{\mathrm{dark}}$ represents the dark reference. The white reference was obtained by capturing a standard white reference tile just before the HS intraoperative acquisitions, keeping the same illumination conditions. The dark reference was acquired by keeping the camera shutter closed. As indicated in [25], the low- and high-frequency bands showed high noise generated by the low performance of the CCD sensor in these extreme bands. For this reason, Bands 1–55 (from 400–440 nm) and Bands 700–826 (from 902–1000 nm) were removed, obtaining HS cubes with 645 spectral bands covering the range comprised between 440 and 902 nm. However, the large number of spectral bands leads to a high-dimensional dataset with high redundancy, which can cause a high computation cost and a decrease of the classification accuracy.

In order to reduce dimensionality but preserve the most relevant spectral information, we applied a band-selection algorithm to optimally select a subset of bands. The objective of band selection is to select an optimal set of bands that preserves the discriminative features and achieves a good classification accuracy. Searching strategies based on metaheuristic algorithms are widely adopted to derive the optimal combination of bands [32,33]. These algorithms are often inspired by nature and have satisfactory convergence behavior. Based on the information gain and the spectral curve, Xie et al. [34] proposed a gray wolf optimizer (GWO) framework, which was verified on the Indian Pines and Salinas HS dataset with the aid of support vector machine (SVM) classifier. Gao et al. [33] performed an ant colony optimization (ACO)-based band selection technique, using both supervised and unsupervised objective functions. The ACO-based band-selection experiments and results showed that the classification accuracy on selected bands was higher than the accuracy found when using all the bands [33]. Prior to band selection, a band-calibration step was applied, by calculating the mean value every eight bands from the original bands.

Band selection has the advantage of preserving the spectral information by removing redundant bands. In this study, The ACO band-selection algorithm was employed to choose the bands with the most distinctive information with respect to the multi-class classification accuracy. The ACO algorithm is created to solve the optimization problem by imitating the behavior of real ants in searching for food [35]. When ants are trying to find the food source, pheromones are released on the route. The shorter the road, the more pheromones will be released. This chemical substance contains information which can attract other ants, so there are always more ants on the route with a higher concentration of pheromones. The pheromone also has an evaporation coefficient, and it gradually vanishes on the route where there are few ants. Due to the positive feedback, the ants can find the shortest route to the food source after a short enough time. In the ACO band-selection algorithm, the artificial ants pass a route that contains *n* different vertices of the graph. The number of vertices represents the chosen bands. During one iteration, the pheromone would be updated after all the ants find a route. The pheromone updating rule can be written as [33]:(2)$$\tau_{ij}=\rho \cdot \tau_{ij}+\Delta \tau_{ij}\;,$$
(3)$$\Delta \tau_{ij}=\left\{\begin{array}{cc} Q\cdot F(f_{\mathrm{best}}) & \upsilon_{i},\upsilon_{j}\in route_{\mathrm{best}}\;,\\ 0 & \upsilon_{i},\upsilon_{j}\notin route_{\mathrm{best}}\;, \end{array}\right.$$
where ρ is the evaporation coefficient, $\Delta \tau_{ij}$ is the increasing pheromone, *Q* is a constant and *F* is a function that can control the pheromone according to the value of objective function *f*. A better $f_{\mathrm{best}}$ denotes a higher $\Delta \tau_{ij}$. The object function is set to the subset accuracy of a linear SVM classifier to perform supervised band selection.

### 2.4. HS Image Classification: A Deep Spectral-Spatial Approach

Traditional classification algorithms (e.g., spectral unmixing, SAM, SVM) have been extensively studied for HSI classification [9]. Recently, convolutional neural networks (CNNs) have been gaining attention in the field of HSI analysis. Despite their efficiency in feature extraction, 2D and 3D CNNs blur the relationship between spectral and spatial dimensions in the HS data. The 2D CNN alone is unable to extract good discriminating feature maps from the spectral dimensions [30]. Unfortunately, 3D CNNs are more complex architectures, which do not show discriminating power for data with similarity over many spectral bands [30]. Combining 3D- and 2D CNNs may overcome the shortcomings of the single models and extract both spatial and spectral information. Paoletti et al. [36] proposed a spectral–spatial capsule network to learn the HS features, while Fang et al. [37] introduced deep hashing neural networks for HSI feature extraction Recently, Roy et al. [30] and Luo et al. [38] proposed the concatenation of 3D convolution and 2D convolution operations to obtain both spatial and spectral information and reduce the complexity of the 3D CNN model. Inspired by the joint assembly of 3D and 2D CNNs and the strategy to achieve a maximum possible accuracy with a hybrid CNN model, we propose the model architecture shown in Figure 2, which is a deeper version of the network proposed by [30]. As input for the network, Roy et al. reduced the number of spectral bands to remove the spectral redundancy with principal component analysis (PCA). PCA is a band extraction method that generates a new set of linearly uncorrelated features where the first few contain most of the original-signal variation [39]. However, the linear projection transformation leads to mixing the original spectral information [39]. As input for our network a band selection algorithm was applied, with the advantage of selecting a subset of bands, preserving their physical meaning and the spatial information. The HybridSN by Roy et al. consists of three 3D convolutional layers, one 2D convolutional layer, three dense layers and a softmax classifier. Our hybrid CNN architecture contains four 3D convolutional layers, two 2D convolutional layers and three fully connected layers where the last layer is a softmax layer. The added convolutional layers may extract more features and improve the classification accuracy in our dataset. Between 3D and 2D convolutions, a reshape layer converts 5D feature vectors to 4D feature vectors. It can be noticed that an HS patch has the shape of P×P×100 where *P* is the patch size and 100 is the number of bands. Using a 2D CNN model to perform the HSI classification, the input patch data will be converted to a P×(P×100) vector, subsequently mixing the spatial and spectral information. The 2D CNN model is not able to perform spectral-feature learning. With a deep 3D CNN model, the computation complexity increases drastically, and the classification performance is not satisfactory for classes that have similar spectral signatures [30]. The hybrid architecture combines the advantages of both 2D and 3D CNNs, keeping the discriminating spectral-spatial features with lower computation complexity. The 3D convolutional layers extract the spectral information, while the 2D convolutional layers learn the distinctive spatial features of different spectral losses. Furthermore, the band selection combined with our low complex model would strengthen the efficiency of using CNN extracting spatial and spectral information in HS data [40]. At last, a softmax activation function is used to give the four-class classification results. The results from the above-described model were benchmarked with the results obtained by using a 2D CNN with three 2D convolutions, one average pooling, and three fully-connected layers. A patch-based classification was performed, dividing the HS cubes into non-overlapping 3D patches. The frequency of each class was counted within every patch and the patch label was assigned based on the majority class within the patch. Non-overlapping 3D patches of size 11×11×100 were created from each HS image. The 11×11 2D patch dimension was empirically chosen as the one achieving higher classification accuracy, after having conducted experiments with several patch sizes (e.g., 6×6 and 22×22).

### 2.5. Traditional Supervised Classification Methods

The results of the proposed framework were compared with traditional supervised classification techniques. The first alternative was to replace the softmax function with a linear SVM classifier. This means that a transfer learning approach was applied, in order to use a CNN as feature extractor and train an SVM classifier. The pre-trained hybrid model was used, and the fully connected layers were removed. The output of the last convolutional layer was then used as an input for the SVM classifier. After the training and validation phase, the results were compared with the previously found results obtained by using the softmax function, and a linear kernel-based SVM model, as shown in Figure 3.

## 3. Experiments

### 3.1. Evaluation Protocols

The four-class classification of normal tissue (NT), tumor tissue (TT), blood vessels (BV) and background (BG) was validated by performing an inter-patient classification. Leave-One-Patient-Out Cross-Validation (LOPOCV) was employed to avoid the double usage of the same patient. To evaluate the performances of each approach, the overall accuracy, per-class sensitivity, and specificity metrics were calculated. Equation (Equation 4) defines the accuracy, where TP denotes the number of true positives, TN is the number of true negatives, FP is the number of false positives, and FN the number of false negatives. Sensitivity and specificity are the true positive rate and true negative rate, respectively. They are defined in the same order by Equations (Equation 5) and (Equation 6):(4)$$Accuracy=\frac{TP+TN}{TP+TN+FP+FN}$$
(5)$$Sensitivity=\frac{TP}{TP+FN}$$
(6)$$Specificity=\frac{TN}{TN+FP}.$$

In addition, the receiver operating characteristic (ROC) curve was computed and the area under the curve (AUC) metric was provided for each class in the results, in one-vs-all way. For each image, a ground-truth map with the labeled pixels of the four classes (NT, TT, HT, and BG) is used for computing the above defined performance metrics.

### 3.2. Results

The proposed framework was evaluated on twelve HS images from nine patients with GBM tumor. The ACO band-selection algorithm required almost three hours of computations by using the subset accuracy of SVM as object function for selecting 100 bands, covering the spectral ranges of 410–423 nm, 457–485 nm, 500–533 nm, 593–621 nm, 638–667 nm, 731–740 nm, 757–771 nm and 802–824 nm. The identified spectral ranges approximately correspond to the ranges found in [29], where an extensive analysis was made for choosing the best ranges in twenty-six HS images, obtained from sixteen adult patients from the same in vivo brain database. In our study, the ACO band-selection algorithm was applied by using the following parameters (described in Section 2.3), chosen as in the original paper [33]: ant count = 30, generation = 30, α = 1.0, β = 10.0, ρ = 0.5, *q* = 10 and strategy = 2.

In order to evaluate our classification model, a four-class classification was performed to test the 3D–2D hybrid network against traditional deep learning and machine learning algorithms. The overall per-class accuracy, sensitivity, specificity and AUC of the four-class classification obtained with the 3D–2D CNN are shown in Table 2. The proposed approach is compared with a 2D CNN and with conventional approaches such as SVM and with the combination of SVM when spatial-spectral features are extracted from the pre-trained 3D–2D CNN, as reported in Table 2. The extended results for each patient and each approach are included in the Appendix A as Table A1, Table A2, Table A3 and Table A4. It should be noted that data distribution is not uniform for all cases, so that any value is applicable for tumor and healthy tissue sensitivity, specificity and AUC in patients 010-03, 014-01, 016-04, 016-05 and 017-01. The 3D–2D hybrid CNN model achieves a mean accuracy of 80% which is 8% higher compared to the accuracy reached with the 2D CNN-based model, implying that more discriminative features are extracted from the 3D–2D hybrid CNN model. The average accuracy of the 3D–2D hybrid CNN combined with the SVM classifier (3D–2D CNN + SVM) is 75%, showing a similar performance of the SVM model (accuracy equal to 76%). Since the combination of 3D and 2D CNNs performs better than the SVM in preserving features, the softmax classifier is more suitable for our experiments. The 3D–2D hybrid CNN has a lower sensitivity in classifying the tumor tissue (68%), compared to the healthy and the blood vessels classification (76% and 74%, respectively). However, higher sensitivity values are found when using the hybrid model compared to the SVM classifier or the 3D–2D hybrid CNN combined with the SVM model. The tumor tissue sensitivity decreases when the 2D CNN is applied, leading to the conclusion that this is an unsatisfactory model. This means that tumor tissue may be ignored, and the left-behind tumor tissue is potentially risky for the patient, since it could cause the recurrence of cancer. It is noteworthy that the sensitivity of the background is higher than 90% for the hybrid model (3D–2D CNN) and the 3D–2D CNN combined with the SVM (3D–2D CNN + SVM) and more than 70% for the SVM and the 2D CNN. Most of the background samples are classified correctly. For all four methods, the results show high tumor specificity, with values higher than 97%, indicating that all methods have a high confidence in classifying non-tumor tissue. This is crucial to avoid the resection of normal tissue during the surgery.

Overall, the 3D–2D hybrid CNN achieves the best results with a mean accuracy of 80%, sensitivity of 76%, 68%, 74%, 96%, specificity of 87%, 98%, 92%, 87%, and AUC of 78%, 70%, 84%, 91%, for normal, tumor, blood vessels and background, respectively (Table 2). While the obtained AUCs for the four classes with the four methods are quite similar, the AUC for the tumor tissue is higher when the 3D–2D hybrid CNN is combined with the SVM classifier.

For benchmarking, we have compared our approach with the results found after having applied the 1D DNN (Deep Neural Network) by Fabelo et al. [28]. This 1D DNN was conformed by two hidden layers of 28 and 40 nodes, respectively, using the rectified linear unit as an activation function. These results are presented in detail Table A5 from the Appendix A. The comparison of the results for each patient, obtained after the ACO band selection, for the overall accuracy, tumor tissue sensitivity, normal tissue sensitivity and tumor tissue AUC are shown in Figure 4, Figure 5, Figure 6 and Figure 7. Due to non-uniform data distribution, Figure 5, Figure 6 and Figure 7 have some not applicable (n/a) values and missing bars.

Furthermore, for the HS images 020-01 and 025-01, a sensitivity of zero is found for the tumor tissue. On one hand, in case of the 020-01 image, the detection of the tumor area is quite challenging, since the tumor is not clearly visible in the surface of the brain during data capturing, although the neuronavigation system indicated the contrary during surgery. Hence, as can be seen in Figure 8a, the spectral signatures from the labeled tumor (red) and normal (blue) tissue are quite similar. Considering the other tumor spectral signatures found in the database, for example from the images 012-01 (Figure 8c) and 015-01 (Figure 8d), the tumor signature from 020-01 is more similar to the normal tissue signature than the tumor tissue signatures from 012-01 and 015-01 images. The differences found between tumor and normal tissue in the spectral range of 600 to 800 nm in 012-01 and 015-01 are not found in 020-01. This could explain the misclassifications of the tumor samples in 020-01 using the LOPOCV methodology. On the other hand, all the HS brain data employed in this study were acquired at the University Hospital Doctor Negrin of Las Palmas de Gran Canaria (Spain), except for the 025-01 image, which was captured at the University Hospital of Southampton (UK). As shown in Figure 8b, a decrease in the level of reflection from 700 nm is produced in the labeled spectral signatures of this image, which are quite different from the other data in the database. Although a specific acquisition protocol was followed to acquire the data, this HS image seems to suffer from a different illumination condition. One possibility is that some kind of protection element was placed in front of the illumination system of the camera lens, thereby filtering the light in the infrared region. Another possibility could be the use of the surgical lights illuminating the brain surface during the capturing process, which can interfere with the acquisition process. The surgical lights are quite powerful and affect the capturing process by interfering with the halogen light of the system, disrupting the calibration stage. In most cases, the proposed algorithms outperform the 1D CNN for the detection of tumor tissue. Similar results are found for the sensitivity of the normal tissue. One of the reasons is that the 1D CNN exploits only the spectral information with a pixel-based approach, where each pixel has a dimension of 1 × 128. Furthermore, the labeling process highly relies on the pathological examination. The ground truth was generated mostly based on the examination results. However, the excision area was very limited, and the data were not sufficient for a 2D-pixel-wise classification. Therefore, 2D CNN in combination with the 3D CNN, used in this study, jointly exploits both spectral and spatial information, using image patches from each pixel with a dimension of 11 × 11 pixels.

The classification results are visualized in Figure 9 for patients 008-01, 008-02, 010-03, 012-01, showing that the proposed 3D–2D hybrid CNN approach is able to correctly discriminate the tumor area, which overlaps with the ground truth. As suggested by Fabelo et al., the obtained low sensitivities are caused by the relatively lower number of tumor samples in the training set, and by the fact that only pixels with a high certainty of correct class were selected, which causes the sparse distributions visualized in Figure 9 [28].

## 4. Discussion

HSI is a non-invasive, non-ionizing technique already employed for brain tumor detection and other neurosurgical applications [28,39]. In this study, new experiments performed on an in vivo HS brain database are presented [25]. The proposed 3D–2D hybrid CNN approach, shows promising results for achieving the highest multi-class classification results, compared with supervised machine learning as well as conventional 2D CNN and 1D CNN approaches. A sensitivity of 68%, specificity of 98% and AUC of 70% for tumor tissue classification is achieved. The obtained specificity is 10% higher compared to the current state of the art [28], where twenty-six in vivo HS cubes from sixteen different patients with glioblastoma and other type of brain tumors are analyzed. This demonstrates a better identification of the tumor pixels with the joint normal, blood vessels and background classification. In [28] Fabelo et al. proposed a 1D DNN architecture as main classifier. In that work, the training set included patients who underwent craniotomy for other diagnoses than GBM. Only the GBM cases were employed for testing the algorithm and the rest of the cases were used for the healthy and blood vessels classification. The accuracy reached by the 1D DNN was 77%, and the same result was achieved with the 2D CNN and a traditional SVM-based approach. In order to reduce the false positives in the multiclass classification, Fabelo et al. [28] proposed to combine the 1D DNN with the 2D CNN applied to a gray-scale representation of the HS data. However, the tumor classification accuracy did not improve (42%). An accurate tumor detection is crucial to delineate the malignant lesions, thereby reducing the risk for recurrence.

In this study, GBM patients were used to train and test the models and to classify tumor, healthy and blood vessels by performing an interpatient cross-validation. When implementing the 1D DNN architecture, introduced by Fabelo et al., the results show an increase of the overall accuracy by 2%, and a tumor sensitivity of 68%, compared to the 19%, achieved by the 1D DNN [28]. Aiming at identifying brain tumor margins to aid the surgeon during resection, the results are promising and are improved compared to the state-of-the-art for the multiclass classification of the in-vivo human-brain HS dataset. As observed by Fabelo et al., since the sensitivity for the tumor tissue was higher in the binary classification, the false negatives are caused by the presence of blood vessels and background classes [28]. In Figure 9a′,b′,d′ tumor pixels are classified as blood vessels. The specificity found for the tumor classes, is quite high on average, leading to the conclusion that the algorithm correctly identifies the non-tumor tissue. This is an important feature for a future intraoperative tool, ensuring that the resected areas are not normal brain tissue which can positively impact the patient outcome. It should be noted that part of the HS acquisitions were made when the superficial tumor was resected, either when normal tissue was removed to visualize the tumor in deeper layers. This procedure may have affected the area acquired, where bleeding and irrigation fluid can have resulted in misclassifications of the tumor pixels [28]. Although the surface of the brain was cleaned prior to HS image acquisition, blood may still have interfered with the images due to the time required for the acquisition process (≈1 min). This may have caused misclassifications, especially between the blood vessels and the tumor classes. This study also has a number of limitations. First, the gold standard map is based on similarities in the HS cubes which introduces a bias, as these same HS similarities are later exploited by the subsequent classification methods. Other important limitations are the lack of a dense ground truth as well as the absence of clinical evaluations on the tumor boundaries. Furthermore, the non-uniform distribution in some cases my cause a very high background sensitivity, while leading to a loss of generalization. As suggested by Fabelo et al. [28], a negative mask in the background could potentially reduce the effects of the non-uniform data distribution in the ground truth. The large variability between different patients represents a further limitation in this study, mainly due to different acquisitions done in the two clinical centers. This difference can affect the robustness of the method. A simple way to address this would be a preceding system processing step that would identify the patient scans on which the system will not work well, and label those as low-confidence predictions.

Further studies should be conducted to evaluate the influence of resection on the spectral signature of tissues. Furthermore, a snapshot HS acquisition system would allow real-time acquisition, without the constraint of acquiring HS images at certain surgical times. A snapshot HS acquisition system may also increase the accuracy of the proposed CNN method, by adding spatial resolution to the system.

In this study, a visualization map was created to evaluate the ability of the system for detecting and localizing the tumor during neuronavigation. The tumor tissue overlapped with the ground truth and the main mismatches were found for the blood vessels which were sometimes classified as tumor. Tumor tissues have higher metabolism, hyperplasia and are more vascularized compared to the normal tissue [9]. This can cause the mismatched classification of blood vessels as tumor tissue. In this sense, HSI could be combined with spectroscopy to better analyze the biological differences between tumor and blood vessels, as proposed by Lai et al. [41]. Analyzing the spectral properties of hemoglobin on the brain surface, for instance, can also lead to a better understanding of the classification outcome [28]. Further research must be performed regarding to the relation between the relevant wavelengths and the biological properties of the different tissue classes. Particularly more research could be in the identification of blood vessels where scattering measurements could be considered for improving their delineation and differentiation.

The aim of this study was to extend the investigation of novel deep learning approaches to discriminate between tumor, normal, blood vessels in the human brain. These results can serve as a basis for defining tumor margins using HSI.

## 5. Conclusions

This study presents a novel 3D–2D hybrid CNN employed for brain tissue classification on glioblastoma patients, using an intraoperative HSI system. The 3D–2D hybrid CNN shows higher accuracy in detecting tumor, healthy and blood vessels compared with the state-of-the-art approaches. Both spectral and spatial features are jointly revealed by using a patch-based classification obtained with a hybrid network. A comparison study has been conducted to exploit the potential of extracting the most discriminative features. A 2D CNN and two conventional classification methods (the SVM, and the SVM classifier combined with the 3D–2D hybrid CNN for feature extraction) have been used to validate the proposed network. Moreover, the method has been compared with the 1D CNN proposed by Fabelo et al., outperforming it in accuracy and tumor sensitivity. This framework only involves data normalization and calibration as a preprocessing step. While in [28] a preprocessing step was added to filter the noise in the spectral signatures, our framework applies the filtering in the learning process, resulting in a higher sensitivity [28].

While further experiments are warranted to optimize the deep learning algorithm and the multi-class classification, the spectral-spatial approach outperforms traditional deep and machine learning techniques and can serve as a robust basis for a future intraoperative real-time system. While the four-class classification is a challenging task, it is essential to provide surgeons with an easy interpretation of the classification map, in which different structures are visualized. The ability of the proposed framework to detect tumor and improve the surgical outcome should be further studied and extended, employing larger patient datasets. Nevertheless, the results achieved in this study prove the feasibility of tumor identification using HSI. These results can serve as a basis for improving the detection of tumor and surrounded tissue using HSI and may open a future for image-guided neurosurgery applications.

## Figures and Tables

**Figure 1 sensors-20-06955-f001:**
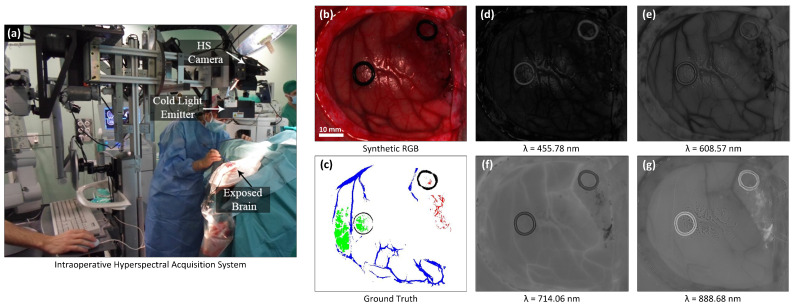
(**a**) HELICoiD intraoperative HS acquisition system (reproduced from [28]). (**b**) Synthetic RGB image of the brain tissue extracted from the hyperspectral cube. The marker has an external diameter of 10 mm. (**c**) Annotated ground truth obtained with the semi-automated Spectral Angle Mapper (SAM) labeling tool. Normal, tumor, blood vessels and background classes are represented in green, red, blue, and black color, respectively. White pixels correspond with non-labeled data. (**d**–**g**) Examples of four gray-scale band representations for different wavelengths in the employed spectral range. These wavelengths were selected by visual inspection of this HS cube in order to illustrate the four of the most different spectral bands.

**Figure 2 sensors-20-06955-f002:**
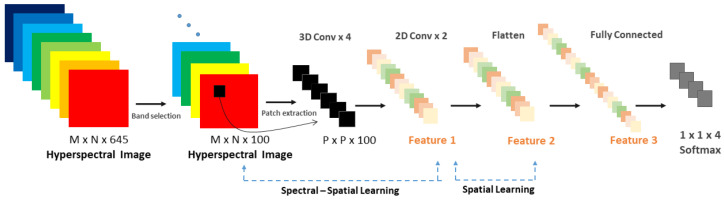
A 3D–2D hybrid Convolutional Neural Network (CNN) architecture for feature extraction and classification adapted from Roy et al. [30]. At the left, the band selection and patch extraction are shown. In the middle, the feature extraction by using 3D and 2D CNNs used for giving the final output depicted at the right.

**Figure 3 sensors-20-06955-f003:**
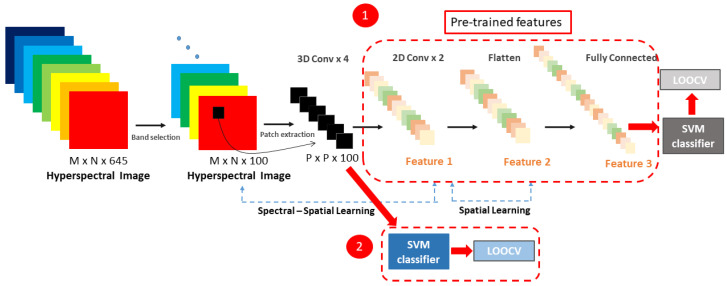
A 3D–2D hybrid CNN architecture for feature extraction and classification adapted from Roy et al. [30]. At the left the band selection and patch extraction is shown. In the middle, (1) feature extraction is shown by using 3D and 2D CNN for giving the final classification output based on an SVM classifier. (2) The intensity values of the patch extracted are used to feed an SVM classifier and perform a supervised classification.

**Figure 4 sensors-20-06955-f004:**
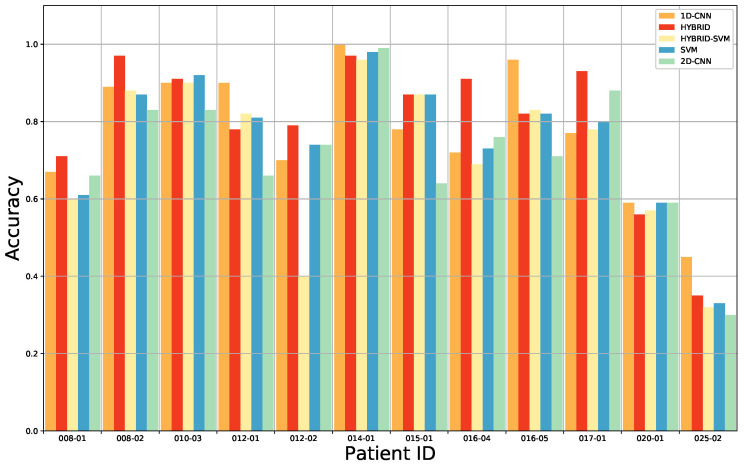
Average accuracy results of the leave-one-patient-out cross-validation of the four-class classification for each method: 1D CNN, 3D–2D hybrid CNN, 3D–2D hybrid CNN + SVM, only SVM and the 2D CNN.

**Figure 5 sensors-20-06955-f005:**
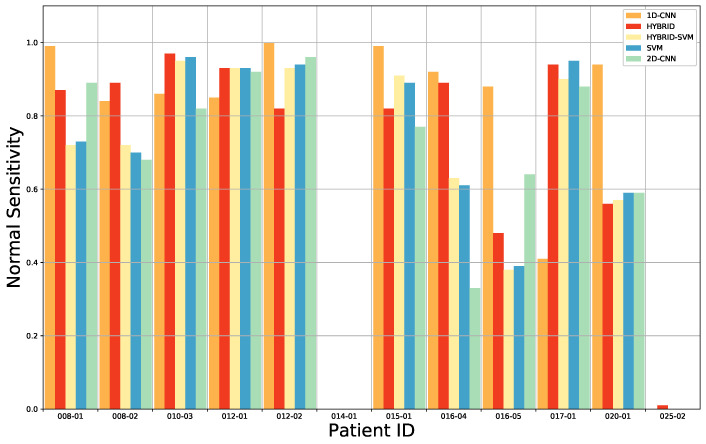
Average sensitivity results for the normal tissue classification of the leave-one-patient-out cross-validation of the four-class classification for each method: 1D CNN, 3D–2D hybrid CNN, 3D–2D hybrid CNN + SVM, the SVM and the 2D CNN.

**Figure 6 sensors-20-06955-f006:**
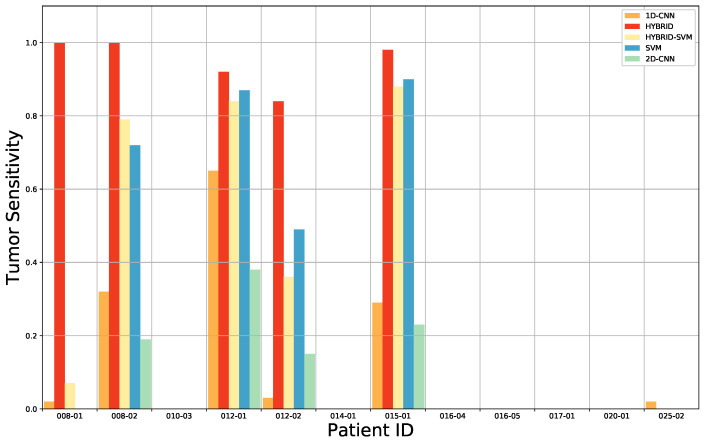
Average sensitivity results for the tumor tissue classification of the leave-one-patient-out cross-validation of the four-class classification for each method: 1D CNN, 3D–2D hybrid CNN, 3D–2D hybrid CNN + SVM, only SVM and the 2D CNN.

**Figure 7 sensors-20-06955-f007:**
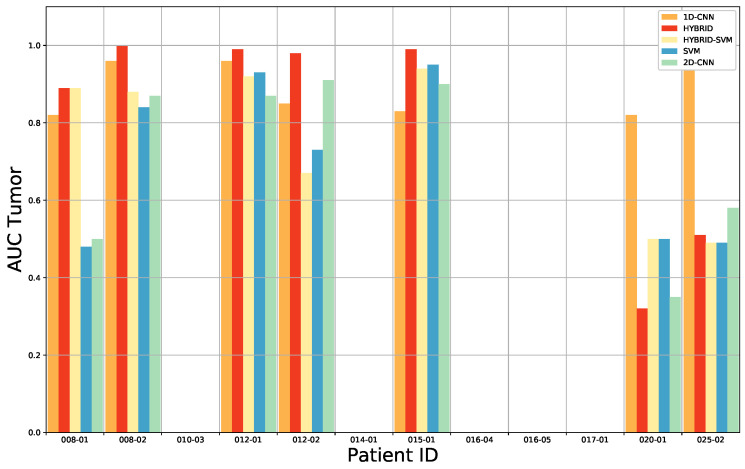
Average AUC results for the tumor tissue classification of the leave-one-patient-out cross-validation of the four-class classification for each method: 1D CNN, 3D–2D hybrid CNN, 3D–2D hybrid CNN + SVM, only SVM and the 2D CNN.

**Figure 8 sensors-20-06955-f008:**
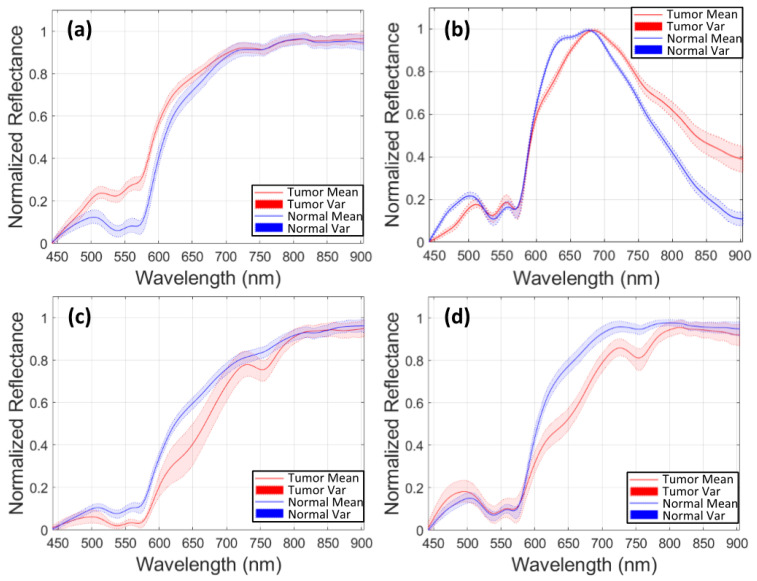
Examples of the different means and variances of the normal (blue) and tumor (red) spectral signatures from different HS images that compose the labeled dataset. (**a**) Data from image 020-01. (**b**) Data from image 025-02. (**c**) Data from image 012-01. (**d**) Data from image 015-01.

**Figure 9 sensors-20-06955-f009:**
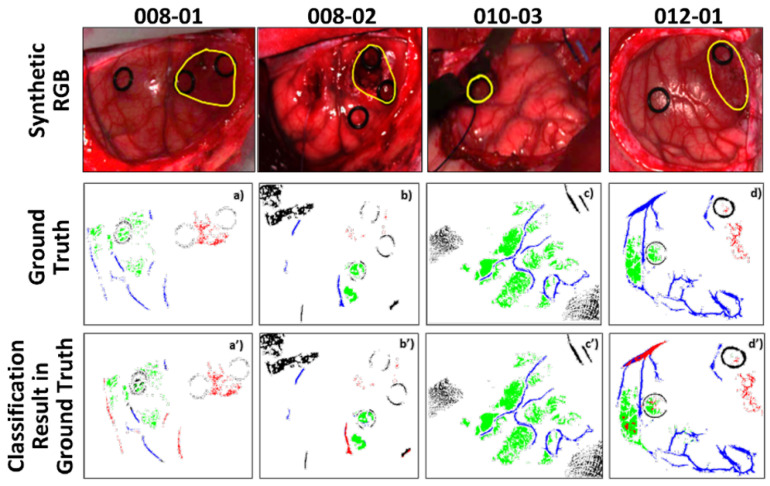
Multi-class classification maps of four of the test hyperspectral (HS) images obtained with the 3D–2D hybrid CNN model. (**a**–**d**) depict the predicted map of their respective ground truth, (**a′**–**d′**) Normal, tumor tissue and blood vessels are represented in green, red, and blue colors, respectively, while the background is represented in black.

**Table 1 sensors-20-06955-t001:** Summary of the hyperspectral (HS) images. NT is normal tissue, TT is tumor tissue, BV is blood vessel, BG is background.

Patient ID-Image ID	Number of Pixels			
	NT	TT	BV	BG
1 = 008-01	2295	1221	1331	630
2 = 008-02	2187	138	1000	7444
3 = 010-03	10,626	0	2332	3972
4 = 012-01	4516	855	8697	1685
5 = 012-02	6553	3139	6041	8731
6 = 014-01	0	30	64	1866
7 = 015-01	1251	2046	4089	696
8 = 016-04	1178	0	1064	956
9 = 016-05	2643	0	452	5125
10 = 017-01	1328	0	68	3069
11 = 020-01	1842	3655	1513	2625
12 = 025-02	977	1282	907	3687
Total	35,396	12,366	27,558	40,486

**Table 2 sensors-20-06955-t002:** Overall results comparison for the approaches evaluated in this study. Accuracy, sensitivity, specificity, and AUC are calculated for each class NT: Normal tissue, TT: Tumor tissue, BV: blood vessels, BG: Background.

Approach		Accuracy	Sensitivity				Specificity				AUC			
			NT	TT	BV	BG	NT	TT	BV	BG	NT	TT	BV	BG
Proposed 3D–2D CNN	Mean	0.80	0.76	0.68	0.74	0.96	0.87	0.98	0.92	0.87	0.78	0.70	0.84	0.91
	Std.D.	0.18	0.28	0.47	0.25	0.04	0.15	0.02	0.08	0.26	0.11	0.21	0.10	0.13
3D–2D-CNN + SVM	Mean	0.75	0.68	0.42	0.73	0.91	0.86	0.98	0.91	0.87	0.81	0.76	0.82	0.91
	Std.D.	0.18	0.30	0.41	0.23	0.09	0.15	0.03	0.08	0.27	0.13	0.20	0.12	0.12
SVM	Mean	0.76	0.70	0.43	0.74	0.93	0.87	0.98	0.92	0.87	0.78	0.70	0.84	0.91
	Std.D.	0.18	0.30	0.42	0.23	0.09	0.15	0.02	0.08	0.26	0.11	0.21	0.10	0.13
2D CNN	Mean	0.72	0.69	0.14	0.77	0.93	0.88	0.97	0.89	0.83	0.88	0.71	0.93	0.93
	Std.D.	0.17	0.29	0.15	0.27	0.08	0.14	0.05	0.12	0.29	0.17	0.23	0.07	0.16
1D DNN	Mean	0.78	0.79	0.19	0.84	0.88	0.94	0.97	0.90	0.82	0.91	0.89	0.89	0.87
	Std.D.	0.16	0.31	0.25	0.29	0.24	0.09	0.05	0.13	0.31	0.23	0.08	0.22	0.29

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
