# Peer review of "Hyperspectral Imaging for Glioblastoma Surgery: Improving Tumor Identification Using a Deep Spectral-Spatial Approach"

_sensors, 2020, doi:10.3390/s20236955_

Round 1

Reviewer 1 Report

Overall

This paper describes research aimed at advancing the application of Hyperspectral Imaging (HSI) in the important field of brain cancer surgery, specifically, Glioblastoma Multiforme (GBM). The challenge is to rapidly extract spectral and spatial information from HSI-derived data in order to distinguish between normal and malignant tissues. Much prior work has been carried out by some of the co-authors of the submitted paper e.g. Ref 15. Fabelo H, et al, “An Intraoperative Visualization System Using Hyperspectral Imaging to Aid in Brain Tumour Delineation.” Sensors 2018, 18, 430; doi:10.3390/s18020430.  The work reported in the present paper therefore builds on this earlier work, as well as that of Roy et al (Ref 32), who proposed a fusion of 3D-2D and convolutional neural network, CNN, termed Hybrid-SpectraNet (HybridSN). The latter was partly substantiated by the current authors when assessing colon cancer margins.

The results obtained in the present study will be of interest to those using hyperspectral imaging in diverse applications.

Specific points

  1. Now, the proposed hybrid deep learning-based framework, using a published in vivo HS brain dataset (Ref 14), appears to go beyond the state-of-the-art by giving higher accuracy for detection of tumour, healthy tissue, and blood vessels. If this is so, it should be emphasised.
  2. The work of Roy et al (Ref 32), proposing a hybrid spectral CNN (HybridSN) for HSI classification, is very significant, and perhaps in the current paper it should be emphasised more and the relationship between Roy et al and the submitted paper further explained; some readers might not see this. Roy et al’s work (Ref 32) appears to have inspired the work in the submitted paper, by the concatenation of 3D convolution and 2D convolution, leading to the model architecture proposed in the paper under review.
  3. (Page 4) The size and distribution of the blood vessels needs to be specified more clearly . You say you have re-defined “hyper-vascularised” as “blood vessels”. It is not clear why you believe that the term “hyper-vascularised” should be confusing for medical readers. The size and vessel density could have some relevance to the choice of wavelength, through consideration of scattering; this needs to be considered.
  4. Page 4 Fig 1. What are the dimensions of the markers? In Fig 1 c, d, e, f, on what basis have the wavelengths been chosen? Page 4 Fig 1b, black is said to indicate background, but there do not appear to be any pixels coded black; is this correct? 
  5. In the present work, technology developed and described in your previous paper, (i.e. the HELICoiD project demonstrator, Ref 15) has been used to collect HIS data. You use the technology described in Ref 15. However, it is important to know if you have modified this technology before using it in the present study. It is possible that the light source was different. Please explain and add a comment in the paper if necessary. Also, can you please justify why the 400 to 1000nm band was used. Possibly because penetration depth is greater in the NIR band. It is possible that a brief description of the HELICoiD project would be useful for some readers.
  6. Section 2.2. Is it the case that this section is reporting the studies carried out under HELICoiD and published in Ref 14? Please clarify. In this section, what were the dimensions of the rubber ring markers?
  7. Again, in section 2.3 please clarify which are new data and which are data collected in HELICoiD, since you do refer to Ref 14 again in the submitted paper.
  8. There could be copyright breach, with Fig 1 (a), part of 1(b), and 1(d) in the submitted paper being the same or similar to Fig 3A, 3B and part of 3D in Ref 14. Permission would be necessary to reproduce such art work and also give acknowledgment. Please investigate this.
  9. Table 1, patient 3 = 010-03. Patient 6 = 014-01, and patients 8, 9, and 10. Please explain the results for NT and TT in these patients.
  10. Section 2.3. Pre-processing is vital to deal with the instrumentation noise; what was the S/N and by how much was it improved? Since you are using the same instrument as used in Ref 14, surely you had similar noise problems and have already dealt with them? You used a band-selection algorithm to help deal with the dimensionality issues; was this not dealt with in Ref 14?
  11. Was the band selection work that you did (use of an ant colony-based (ACO) optimisation) actually done as part of the present study or are you quoting Gao et al (Ref 34) band selection with addition of support vector machine (SVM) ? Please clarify. Also the pheromone updating rule seems to be from Gao et al (Ref 34).
  12. Page 8. Experiments. The evaluation protocols are largely sound and straightforward. However, the number of 12 images from only nine patients with GBM is rather small.
  13. Can you be comfortable in using results from Ref 31 for comparison of spectral ranges, or is this just for comparison purposes?
  14. The four-class classification carried out to test the 3D-2D hybrid network against traditional deep learning and machine learning algorithms. The parameters are from Ref 34, which is a reasonable start. The effects of the non-uniform distribution in some cases (line 236) could benefit from further consideration and discussion.
  15. The lower sensitivity in classifying the tumour tissue (68%) compared to the healthy and the blood vessels classification (76% and 74%) may be seen in context with the result of the 2D CNN, where tumour tissue sensitivity decreases; please discuss. Nevertheless, the overall results for mean accuracy with 3D-2D hybrid CNN are good.
  16. Line 274-5. “the tumour signature from 02001 is more similar to the normal tissue than the tumour tissue”. Is this sentence correct?
  17. The 3D-2D hybrid CNN approach presented here, achieves the highest multi-class classification performance, compared with supervised machine learning and conventional 2D CNN and 1D CNN approaches.

Reviewer 2 Report

  1. The authors proposed a framework consisting of a 3D-2D hybrid CNN-based approacmployh to create a joint extraction of spectral and spatial information from hyperspectral images. A comparison study has been conducted exploiting a 2D CNN, 1D CNN and two conventional classification methods (SVM, and the SVM classifier combined with the 3D-2D hybrid CNN) have been used to validate the proposed network. An overall accuracy of 80% was found when tumor, healthy tissue and blood vessels were classnified, clearly outperforming the state-of-the-art approaches. The authors claimed that the results can serve as a basis for brain tumor classification using HSI, and may open future avenues for image-guided neurosurgical applications. However, All the HS brain data employed in this research seems too few. Is it possible to conduct the experiment with more datasets?
  2. Line 141: "Table 1. Summary of the HS images. NT is normal tissue, TT is tumor tissue, BV is blood vessel, BG is background", it should be put on the top of Table.
  3. Line 300: all of the legends of Table 2, Line 388: Table A1, Table A2, Table A3, Table A4 and Table A5 should be put on the top of Tables.
  4. Line 214: FN' in Eqn(4), (5) and FP' in Eqn(6) should be corrected as FN and FP, respectively.
  5. Line 144: Iwhite "represents" the intensity value........, the author should give a verb (represents) for the sentence.
  6.  

Reviewer 3 Report

The paper is well written; some minor typos (see for example row 77). would be good to see experimental setup schematic or photo in materials and methods.

understanding each HS image has many pixels but 26 images seem a small group of data. Can the author add something about data selection and the use of CNN efficacy? 

Reviewer 4 Report

I greatly enjoyed reading the manuscript "Hyperspectral imaging for glioblastoma surgery: improving tumor identification using a deep spectral-spatial approach."  The paper is detailed, well-referenced and has enough background to be a stand-alone reference for a knowledgeable reader.  The authors' results showing a 10% increase in specificity using the 3D-2D hybrid CNN method is impressive.  Being able to identify tumor margins during the operation seems like a ground breaking advancement to me.  I recommend the paper be published as-is.

In future work, it would be interesting to see if enhanced specificity could be achieved by moving to the NIR or SWIR regions, as the QTH lamp can provide illumination in these wavelength ranges.  The SWIR especially since it can detect overtone bands of some materials/chemicals.
